# Improved security bound for the round-robin-differential-phase-shift quantum key distribution

Zhen-Qiang Yin[1,2,3], Shuang Wang[1,2,3], Wei Chen[1,2,3], Yun-Guang Han[1,2,3], Rong Wang[1,2,3], Guang-Can Guo[1,2,3] & Zheng-Fu Han[1,2,3]

The round-robin-differential-phase-shift (RRDPS) quantum key distribution (QKD) protocol has attracted intensive study due to its distinct security characteristics; e.g., information leakage is bounded without learning the error rate of key bits. Nevertheless, its practicality and performance are still not satisfactory. Here, by observing the phase randomization of the encoding states and its connection with eavesdropper's attack, we develop an improved bound on information leakage. Interestingly, our theory is especially useful for implementations with short trains of pulses, and running without monitoring signal disturbance is still available. As a result, the practicality and performance of RRDPS are improved. Furthermore, we realize a proof-of-principle experiment with up to 140 km of fiber, which has been the longest achievable distance of RRDPS until now, whereas the original theory predicted that no secret key could be generated in our experiment. Our results will help in bringing practical RRDPS closer to practical implementations.

---

[1] CAS Key Laboratory of Quantum Information, University of Science and Technology of China, Hefei 230026, China. [2] Synergetic Innovation Center of Quantum Information & Quantum Physics, University of Science and Technology of China, Hefei, Anhui 230026, China. [3] State Key Laboratory of Cryptology, P.O. Box 5159, Beijing 100878, China. Correspondence and requests for materials should be addressed to S.W. (email: wshuang@ustc.edu.cn) or to W.C. (email: weich@ustc.edu.cn)

Unlike public-key cryptography, whose security relies on unproven mathematical assumptions, quantum key distribution (QKD)[1,2] can information-theoretically distribute secret key bits between distant peers (such as Alice and Bob). According to quantum mechanics, any eavesdropping on a quantum channel will inevitably introduce signal disturbance, which implies that Alice and Bob can bound the information leakage for the eavesdropper (Eve) by collecting the error rate of their raw key bits or some other parameters reflecting the signal disturbance. For the well-known BB84[1] and measurement-device-independent (MDI)[3] QKD with decoy states[4–6], the error rate and counting yields are used to evaluate Eve's information. In coherent-one-way (COW)[7,8] and differential-phase-shift (DPS)[9,10] protocols, the visibility of interference plays an essential role in monitoring information leakage. Device-independent (DI)[11–13] QKD relies on monitoring the violation of Bell inequalities. MDI-QKD and DI-QKD feature a high security level in practice, while COW and DPS have compact and simple implementation. There has been great progress on experimental QKD, such as long-distance QKD implementations[8,14,15], high key rate systems[16–19] and demonstrations of QKD networks[20–23]. Nevertheless, signal disturbance monitoring is indispensable for almost all these QKD protocols and implementations.

Surprisingly, the recently proposed round-robin-differential-phase-shift (RRDPS)[24] protocol is an exception. In the RRDPS protocol, Alice prepares a series of pulse trains, each consisting of $L$ weak coherent pulses. The pulses are individually modulated to random phases out of 0 and $\pi$, and every $L$-pulse train can be handled as a packet. Upon receiving these packets, Bob measures the phase shift between the $m$-th pulse and the $(m + r)$-th pulse of each packet, where $r$ is randomly chosen from $[1, L - 1]$ for each packet and $m + r \leqslant L$. Through a simple and comprehensive security proof[24], it has been noted that Eve's information on raw key bits $I_{AE}$ is no larger than $h_2(N/(L - 1))$, where $N$ is the photon-number of a packet. The main merit of the RRDPS protocol is that the estimation of $I_{AE}$ does not depend on the error rate of key bits, and thus can be treated as a constant experimentally, which implies that signal disturbance monitoring can be removed during the RRDPS protocol.

There are several reasons for the strong interest in RRDPS. In theory, the result sheds new light on how intrinsic randomness of quantum mechanics can be related to secure key distribution. In practice, the removal of signal disturbance monitoring means that we do not need to consider the statistical fluctuations in the monitoring error rate and some other parameters, so a better tolerance of finite-sized-key effects is expected[24]. In particular, the finite-sized-key effects must be carefully considered in practice, since the fluctuations induced by environmental disturbance will lead to inaccurate statistical results or much more consumption for sampling. From the point of view of QKD engineering, the post-processing of QKD can be simplified too, since the random sampling and classical authenticated communications necessary in monitoring signal disturbance are not needed. Furthermore, according to the formula $I_{AE} \leqslant h_2(N/(L - 1))$, it is obvious that the information leakage will be deeply suppressed, and a positive key rate under a higher error rate is expected when $L$ becomes larger, which is the reason why, at the present stage, large $L$ is important. It is worth noting that multi-dimensional QKD protocols[25] usually have higher tolerance of error rate; in particular, the recently proposed Chau15 protocol[26] can tolerate an error rate of up to 50% in principle, but these protocols must run with signal disturbance monitoring.

There have been several successful demonstrations of RRDPS with passive interferometers[27,28] and actively selectable components[29,30]. The longest achieved distance is ~90 km[30]. Despite these experimental successes, it is still a great challenge to realize a practical measurement system with a large $L$ value. In addition, it should be noted that large $L$ values result in lower secret key rates per pulse. Therefore, an improved estimate of $I_{AE}$ would be very useful, especially if it could allow operation with few pulses. In addition, although the upper bound of $I_{AE}$ given in ref. [24] does not depend on the error rate, it is still not fully clear how Eve's attack introduces error bits and if it is possible to use the error rate in RRDPS to improve its performance. To address these issues, we report an improved theoretical bound on $I_{AE}$. Interestingly, error rate can also be considered in our method to estimate $I_{AE}$ in an even tighter way. Through numerical simulation, we show that with our theory, the performance of the real-life RRDPS implementation can be improved. Even the RRDPS with relatively small $L$, e.g., $L = 8$, can outperform commonly used BB84 with decoy states when interferometer misalignment is severe. It is also remarkable that the RRDPS protocol with $L = 3$, which is not permitted in the original RRDPS protocol, can generate secret keys according to our analysis. Finally, we verify our theory through a proof-of-principle experiment with $L = 3$, which can run at a distance of 30 km without signal disturbance monitoring and decoy states. In addition, a demonstration at 140 km is also realized with monitoring signal disturbance and decoy states.

## Results

**New bound for Eve's information.** The original security proof given in ref. [24] is simple and elegant but does not consider Eve's optimal attack and corresponding information leakage. Our basic idea is to directly construct Eve's collective attack to each packet and calculate the maximal information acquired by Eve. Considering that the quantum de Finetti theorem[31–33] holds when the pulses are grouped by packets in RRDPS, the results also then cover general coherent attacks. However, even in case of collective attack, it is not easy to perform theoretical analysis, since the dimension of Alice's encoding state depends on $L$ and may be very large. For simplicity, we first consider the case in which each packet contains only one photon. Alice randomly prepares the single-photon state $|\psi\rangle = \sum_{m=1}^{L}(-1)^{k_m}|m\rangle$, where $k_m \in \{0, 1\}$ is Alice's raw key bit and $|m\rangle$ ($m \in \{1, .., L\}$) indicates that this single-photon is in the $m$-th time-bin. Eve's general collective attack can be given by $U_{Eve}|m\rangle|e_{initial}\rangle = \sum_{n=1}^{L} c_{mn}|n\rangle|e_{mn}\rangle$, where the quantum state of Eve's ancilla $|e_{mn}\rangle$ corresponds to Eve transforming $|m\rangle$ to $|n\rangle$ and sending $|n\rangle$ to Bob. In principle, Eve's ancilla has $L^2$ different states and thus is difficult to work with. We develop a method to simplify Eve's quantum state and bound her information effectively. The essence of our method is to introduce phase randomization, which was not considered in previous works. Concretely, consider the case in which Bob has measured the incoming single-photon with basis $|a\rangle \pm |b\rangle$ successfully and announced $(a, b)$ publicly. Eve then aims to guess $k_a + k_b$. For any $m \neq a, b$ the phase $(-1)^{k_m}$ is completely random to Eve, which implies that some mixed components $|c_{ma}|^2|e_{ma}\rangle\langle e_{ma}| + |c_{mb}|^2|e_{mb}\rangle\langle e_{mb}|$ ($m \neq a, b$) will emerge in the density matrix of Eve. These mixed components do not give Eve any information and can thus be ignored to simplify the proof. Accordingly, we find that $I_{AE} \leqslant \max_{0 \leqslant x \leqslant 1} \varphi((L-1)x, 1-x)/(L-1)$, where $\varphi(x, y) = -x\log_2 x - y\log_2 y + (x + y)\log_2(x + y)$. In addition, $x$ can be related to the error rate $E$, so this bound works for implementations both with and without monitoring signal disturbance. One can refer to Supplementary Note 1 for the detailed security proof.

It would be very useful to extend the security proof from the single-photon to the $N$-photon case. Nevertheless, due to the complexity of the $N$-photon quantum state, it is difficult to depict and estimate the upper bound of Eve's information for the general $N$-photon case. Our technique is based on grouping the

$N$-photon state into different summations with different numbers of phases and introduce phase randomization between them. Here, we sketch our method for the odd-$N$ photon-numbers case. Such an odd-$N$ ($N \leqslant L-1$) photons quantum state must have the form $|\psi\rangle = \sum_{t=1}^{N/2+1/2} (-1)^{k_{m_1}+...+k_{m_{2t-1}}}|m_1 m_2 ... m_{2t-1}\rangle_{\text{odd}}$, in which $|m_1 m_2 ... m_{2t-1}\rangle_{\text{odd}}$ means a superposition of quantum states in which the photon-numbers in time-bins $m_1 m_2 ... m_{2t-1}$ must be odd, while the photon-numbers in all other time-bins must be even. The form of $|m_1 m_2 ... m_{2t-1}\rangle_{\text{odd}}$ depends on the values of $N$ and $L$. For example, assuming $N=3$ and $L=5$, non-normalized state $|1\rangle_{\text{odd}} \triangleq \sqrt{3}|1\rangle(|2\rangle|2\rangle + |3\rangle|3\rangle + |4\rangle|4\rangle + |5\rangle|5\rangle) + |1\rangle|1\rangle|1\rangle$, where $|1\rangle|2\rangle|2\rangle$ means that there is one photon in the first time-bin and two photons in the second time-bin, while $|1\rangle|1\rangle|1\rangle$ indicates that all three photons occupy the first time-bin (see Supplementary Note 3 for the three-photon case). It is then straightforward to redefine the collective attack with the new basis $|m_1 m_2 ... m_{2t-1}\rangle_{\text{odd}}$: $U_{\text{Eve}}|m_1 m_2 ... m_{2t-1}\rangle_{\text{odd}}|e_{\text{initial}}\rangle$ $= \sum_{n=1}^{L} c_{m_1 m_2 ... m_{2t-1} n}|n\rangle|e_{m_1 m_2 ... m_{2t-1} n}\rangle$, where the quantum state of Eve's ancilla $|e_{m_1 m_2 ... m_{2t-1} n}\rangle$ corresponds to Eve transforming $|m_1 m_2 ... m_{2t-1} n\rangle_{\text{odd}}$ to single-photon state $|n\rangle$ and sending $|n\rangle$ to Bob. After Bob measures the incoming single-photon with basis $|a\rangle \pm |b\rangle$ and announces $(a, b)$ publicly, Eve will try to guess $k_a + k_b$. Due to the potential phase randomization between different summations, Eve can acquire some information only from two types of "two-dimensional" terms such as $U_{\text{Eve}}(-1)^{k_{m_1}+k_{m_2}+...+k_{m_{2t-1}}}((-1)^{k_a}|m_1 m_2 .. m_{2t-1} a\rangle_{\text{odd}}$ $+ (-1)^{k_b}|m_1 m_2 ... m_{2t-1} b\rangle_{\text{odd}})|e_{\text{initial}}\rangle$ and $U_{\text{Eve}}(-1)^{k_{m_1}+k_{m_2}+...+k_{m_{2t-1}}}(|m_1 m_2 ... m_{2t-1}\rangle_{\text{odd}}$ $+ (-1)^{k_a+k_b}|m_1 m_2 ... m_{2t-1} ab\rangle_{\text{odd}})|e_{\text{initial}}\rangle$ ($m_1, m_2, ..., m_{2t-1} \neq a, b$). Summing over the upper bounds of Eve's information on all these "two-dimensional" terms, we obtain the final formula to estimate Eve's information. The examples for two-photon and four-photon cases are given in Supplementary Notes 2 and 4, respectively. The detailed proof for general cases can be found in Supplementary Notes 5–7. The results are summarized by the following theorem and its corollary.

**Theorem.** For the RRDPS protocol with L-pulse packet, where each packet contains N photons ($L \geqslant N+1$), Eve's information can be bounded by

$$I_{\text{AE}} \leqslant I_{\text{AE}}^{\text{U}} \triangleq \text{Max}_{x_1, x_2, ..., x_{N+1}} \left\{ \frac{\sum_{n=1}^{N} \varphi((L-n)x_n, nx_{n+1})}{L-1} \right\}, \quad (1)$$

where $\varphi(x, y) = -x \log_2 x - y \log_2 y + (x+y)\log_2(x+y)$, non-negative real parameters $x_i$ satisfying $\sum_{i=1}^{N+1} x_i = 1$. Moreover, if the error rate of raw key bits is E, these parameters $x_i$ ($i \in \{1, 2, ..., N+1\}$) must satisfy the constraint

$$E \geqslant \frac{\sum_{n \geqslant 1}^{(N-1)/2} \left(\sqrt{(L-2n)x_{2n}} - \sqrt{2nx_{2n+1}}\right)^2 + (L-N-1)x_{N+1}/2}{L-1}$$

for odd $N$,

$$(2)$$

and,

$$E \geqslant \frac{\sum_{n \geqslant 1}^{N/2} \left(\sqrt{(L-2n+1)x_{2n-1}} - \sqrt{(2n-1)x_{2n}}\right)^2 + (L-N-1)x_{N+1}/2}{L-1}$$

for even $N$.

$$(3)$$

**Table 1 $E_{\text{max}}$ of RRDPS with different methods**

| L \ Method | Original RRDPS | Eq. (1) without E | Eq. (1) with E |
|---|---|---|---|
| 3 | – | 0.0546 | 0.0811 |
| 5 | 0.0289 | 0.122 | 0.144 |
| 8 | 0.0818 | 0.176 | 0.191 |
| 16 | 0.165 | 0.244 | 0.252 |
| 32 | 0.24 | 0.3 | 0.303 |

**Corollary.** If the photon-number $L \leqslant N-2$, $I_{\text{AE}} < 1$ always holds.

Based on this theorem, the upper bound of $I_{\text{AE}}$ is generalized to find the maximum value of a given function under a constraint defined by $E$. If we ignore this constraint, we obtain $I_{\text{AE}}^{\text{U}}$ without monitoring signal disturbance. Alternatively, if we retain this constraint, a tighter estimation may be achieved. It is remarkable that searching such a maximum value can be effective and concise through a numerical method, since its function is convex. A different, less tight bound improvement was recently reported in ref. [34].

**Potential improvements made by our theory.** For a QKD protocol, the mutual information between Alice and Bob is given by $I_{\text{AB}} = 1 - h_2(E)$. Thus, there is a maximum value $E_{\text{max}}$ of error rate $E$, which satisfies $I_{\text{AB}} = I_{\text{AE}}^{\text{U}}$, when $E = E_{\text{max}}$. Obviously, if $E \geqslant E_{\text{max}}$ holds, $I_{\text{AB}}$ will be no larger than $I_{\text{AE}}^{\text{U}}$, and no secret key bits can be generated. Thus, $E_{\text{max}}$ is the maximum value of tolerable error rate of a QKD protocol. We first compare $E_{\text{max}}$ of RRDPS between the original method and our new formulae. In Table 1, we list the results for the cases in which a single-photon source is equipped. One can see that our formulae can increase $E_{\text{max}}$, especially when $L$ is small. It is remarkable to note that for the case $L = 3$, with our formulae $E_{\text{max}}$ can be up to 5%, while the original RRDPS protocol cannot generate secure key bits at all. More importantly, the notable difference between columns two and three in Table 1 implies that our theory leads to increased $E_{\text{max}}$ compared to the original RRDPS even when signal disturbance monitoring is still turned off.

The most important step is to evaluate the secret key rate and achievable distance of RRDPS through simulations. Here, we assume the pulse width is constant for different values of $L$ and the dark counting rate of the single-photon detector (SPD) is set to be $d = 10^{-6}$ per pulse, which is typical and practical. Another important parameter used in the simulations is the interferometer misalignment $e_{\text{mis}}$. In a phase-coding system, the error rate mainly stems from apparatus imperfections, such as interferometer misalignment and dark counts of SPD. The interferometer misalignment $e_{\text{mis}}$ indicates the probability that an incoming photon hits an erroneous SPD due to interferometer misalignment. Actually, $e_{\text{mis}}$ depends on the visibility $V$ of optical interference, and $e_{\text{mis}} = (1 - V)/2$. In an ideal interferometer with $V = 1$, two optical pulses with relative phases 0 and $\pi$ always hit different SPDs. Thus, one can deduce the relative phase, i.e., key bit in the phase coding system, by observing which SPD clicks. However, due to the limited precision of interferometer fabrication or the environmental disturbance, e.g., the drift between the lengths of the short arm and long arm of the interferometer, $V$ may be lowered, and higher $e_{\text{mis}}$ is introduced.

Although $e_{\text{mis}}$ in most reported QKD experiments can be kept small, i.e., $e_{\text{mis}} \leqslant 5\%$, it is still important to evaluate the performance of RRDPS in the high $e_{\text{mis}}$ region. There are two reasons for this. First, reducing $e_{\text{mis}}$ requires complex techniques, such as active feedback[35] and interferometer fabrication with high precision. In addition, active feedback techniques, such as phase-reference alignments and polarization controls, may be ineffective

**Table 2 loss$_{max}$ (dB) of BB84 and RRDPS with different methods**

| $e_{mis}$ \ Method | BB84 | Original RRDPS | Eq. (1) without $E$ | Eq. (1) with $E$ |
|---|---|---|---|---|
| 0.015 | 47.8 dB | 41.5 dB ($L = 19$) | 44.9 dB ($L = 7$) | 45.6 dB ($L = 7$) |
| 0.08 | 42.8 dB | 39.8 dB ($L = 29$) | 42.7 dB ($L = 8$) | 43.5 dB ($L = 8$) |
| 0.1 | 38 dB | 39.2 dB ($L = 30$) | 42 dB ($L = 9$) | 42.8 dB ($L = 9$) |
| 0.15 | – | 37.7 dB ($L = 45$) | 40.6 dB ($L = 16$) | 41 dB ($L = 16$) |

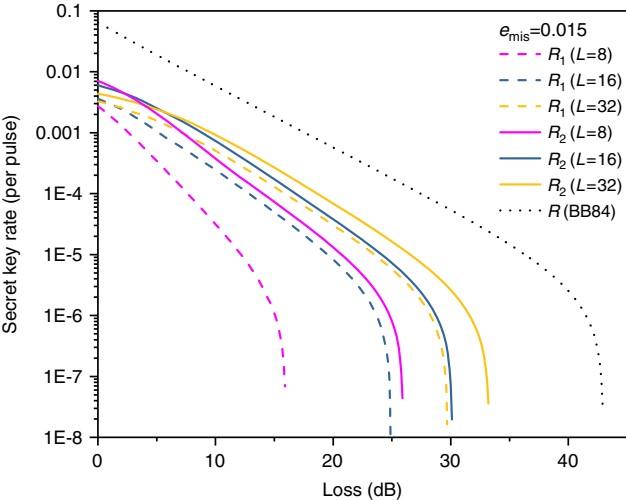

**Fig. 1** Secret key rate $R$ versus channel loss. $R_1$ and $R_2$ represent the original RRDPS protocol and the proposed one, respectively. $R$ (BB84) is for the BB84 protocol with infinite decoy states. Both $R_1$ and $R_2$ are simulated for the scenarios without monitoring signal disturbance

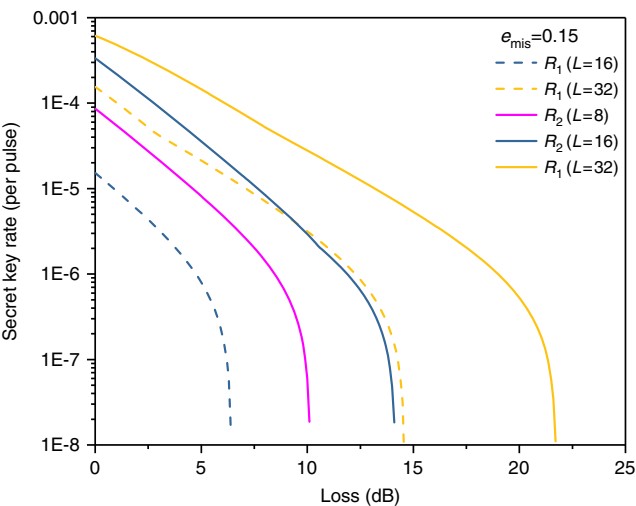

**Fig. 2** Secret key rate $R$ versus channel loss. $R_1$ and $R_2$ represent the original RRDPS protocol and the proposed one, respectively. The line for BB84 is not drawn since its key rate is always 0 in this case. Both $R_1$ and $R_2$ are simulated for the scenarios without monitoring signal disturbance

and even invalid in fast-changing environments. High-precision interferometers, on the other hand, are more challenging in RRDPS, since interferometers with variable delays must be matched well. To improve the robustness of the QKD system in various environments and alleviate its dependence on these techniques, QKD protocols inherently feasible in high $e_{mis}$ scenarios are highly desirable. Second, the use of other high-dimensional degrees of freedom, e.g., orbital angular momentum (OAM) of photons, rather than time-bin, is a potential way to improve the key rate of RRDPS, but typically, $e_{mis}$ in QKD based on OAM can be greater than 10%[36,37]. Hence, simulations of RRDPS with large $e_{mis}$ are relevant for future study. We here report the simulation results. Details of how we model the QKD systems and perform the simulations are presented in the methods section.

For ease of understanding, we first study the tolerance of channel loss when a single-photon source is equipped. Under a given $e_{mis}$, secure key bits can be generated only when the channel loss is smaller than a value loss$_{max}$, which is understood as the maximum value of tolerable channel loss. The loss$_{max}$ values under different $e_{mis}$ are listed in Table 2. For RRDPS, the value of $L$ is optimized to maximize loss$_{max}$. We can see that with the help of formula (1), the loss$_{max}$ of RRDPS becomes much larger, and $L$ can be lowered, compared to the original RRDPS. When $e_{mis} \geqslant 0.08$, the improved RRDPS can outperform BB84 in terms of tolerable channel loss.

Next, we consider a more realistic scenario in which a weak coherent source is used. The secret key rates $R$ per pulse versus total losses for $L = 8$, $L = 16$ and $L = 32$ are simulated. Figure 1 ($e_{mis} = 0.015$) and Fig. 2 ($e_{mis} = 0.15$) are both simulated without using signal disturbance parameters. From them, we can see that with the help of the proposed method, the secret key rate and

achievable distance of RRDPS systems are both evidently increased, especially for small $L$ cases. To further investigate the performance of our improved RRDPS under high $e_{mis}$, the secret key rates versus $e_{mis}$ under typical channel losses are depicted in Fig. 3, from which we can see that RRDPS outperforms BB84 with decoy states when $e_{mis}$ is greater than 10%.

We also analyzed experimental data from previous experiments to show the improvement of key rate. In the experiment of RRDPS with $L = 65$ reported in ref. [29], the secret key rate for a 95 km fiber channel can be increased from $5 \times 10^{-8}$ to $1.4 \times 10^{-6}$ per pulse (see methods section for details).

These simulations confirm the prominent advantages of our theory over the original RRDPS. Compared with BB84 with decoy states, the proposed RRDPS is inherently feasible under a high $e_{mis}$ region, which is meaningful to alleviate its dependence on implementation techniques, such as active phase compensations and interferometer fabrication with high precision. In addition, the proposed RRDPS can still run without monitoring signal disturbance, so its unique advantages on the tolerance of finite-sized-key effects and postprocessing convenience over conventional QKD protocols are maintained.

**Proof-of-principle experiment**. Based on the above theoretical results, RRDPS requires that $L \geqslant 3$. This would represent the simplest RRDPS experimental implementation up to date. Here, we describe a proof-of-principle experiment with $L = 3$ to verify our theory.

Our implementation is shown in Fig. 4, and it is similar to the ones employed in refs [28,29]. At Alice's site, a pulse train with a repetition rate of 1 GHz is generated by modulating a 1550.12 nm continuous wave (CW) laser using the first LiNbO$_3$ intensity modulator (IM$_1$). Every 3 pulses ($L = 3$) is defined as one packet.

The second intensity modulator (IM$_2$) is employed to implement the decoy states method, by which each packet is randomly modulated into signal, decoy and vacuum packets. The first phase modulator (PM$_1$) adds phase $-\pi/2$ or $\pi/2$ on each pulse to encode the key bits, and the second phase modulator (PM$_2$) adds a random global phase on each packet. The encoded pulse train is then launched into a variable attenuator (VA) so that the average photon-number per pulse can be adjusted.

At Bob's site, the passive scheme based on a $1 \times 2$ beam splitter (BS) is used to implement a high-speed and low-loss decoding measurement. Since $L = 3$ and the time interval between adjacent pulses is 1 ns, there are only two unbalanced Faraday–Michelson interferometers (FMIs) with 1 and 2 ns temporal delays. One 50/50 BS and two Faraday mirrors (FMs) constitute an FMI, and a three-port optical circulator is added before the BS to export the other interference result. Each output of these two unbalanced FMIs is led to an SPD. Finally, the detection events are recorded by a time-to-digital convertor (TDC), which records the time-tagged and which-detector information.

The passive implementation scheme and small values of $L$ make our RRDPS system very practical. The passive approach allows high time efficiency and internal transmittance of Bob's

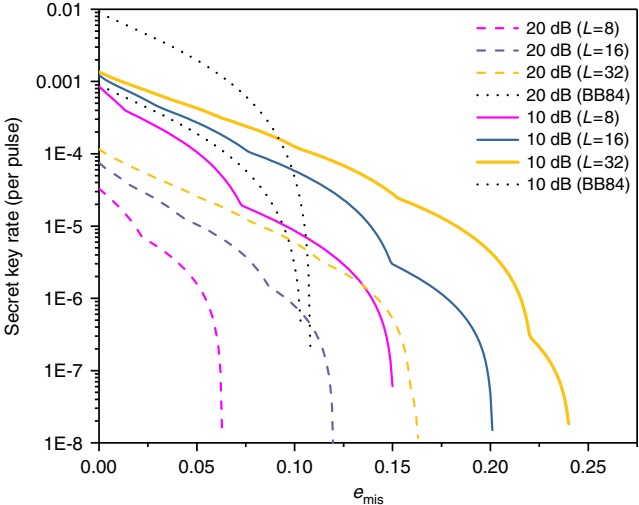

**Fig. 3** Secret key rate $R$ versus $e_{\mathrm{mis}}$. The dashed lines and solid lines represent the proposed RRDPS under channel losses of 20 and 10 dB, respectively. Signal disturbance monitoring is still turned off. The two dotted lines are for BB84 with infinite decoy states under channel losses of 20 and 10 dB, respectively

optical components, and four SPDs used to detect the $L = 3$ packet are acceptable. The $1 \times 2$ BS amounts to randomly choosing between 1 and 2 ns delay FMIs. In contrast to active schemes[29], the passive choice between different delay measurements has no speed limits, and the time interval between each two packets to achieve a low error rate is no longer necessary. The average insertion loss (IL) of the 1 and 2 ns delay FMIs is only approximate 0.80 dB, in which the IL of the optical circulator is also included. These two FMIs are placed in two small ABS plastic cases to isolate them from the environment, and heating plates are used to keep the temperature of FMIs above room temperature. Thus, we could actively and independently compensate for the phase shifts of 1 and 2 ns delay FMIs and keep the phase of the unbalance interferometer stable. Owing to 45° Faraday mirrors, these FMIs are insensitive to polarization variations and feature extinction ratios of approximately 23.5 dB.

In addition, SPDs based on InGaAs/InP avalanche photodiodes (APD) are employed to detect photons from 1 and 2 ns delay FMIs, which makes the RRDPS system more practical. These four SPDs are working with Peltier cooling and operated in gated Geiger mode with the sine-wave filtering method[38]. The detection efficiencies of the four SPDs are approximately 20.4% with a dark count rate of $1.25 \times 10^{-6}$ per gate and an after-pulse probability of 1.02%. Here, the insertion loss of the optical circular from BS (of FMI) to SPD is included in the detection efficiency of SPD.

We tested the $L = 3$ RRDPS system with standard telecom fiber channels at distances of 30, 50, 100, and 140 km. At a distance of 30 km, the system was running without monitoring signal disturbance and decoy states, while the mean photon-number per pulse was set to be 0.005. At the other distances, our RRDPS system cannot generate secret key bits without decoy states due to the contribution of multi-photon events, so decoy states must be introduced. The decoy states method was implemented by setting the photon-numbers per pulse of the signal, decoy, and "vacuum" packets with values of 0.13, 0.03, and 0.0003, respectively. These values of mean photon-number are optimal to maximize the key rate according to the simulations. The experimental results are listed in Table 3, where the error rates of key bits and yields per packet are directly obtained experimentally. When decoy states are employed, we use formulae given in ref. [39] to calculate the yield and error rate for a single-photon packet. The secret key rates $R_1$ and $R_2$ are then calculated according to Eq. (1) without and with the error rate, respectively (see methods section for detailed information).

In our $L = 3$ RRDPS experiment without monitoring and decoy states, the transmission distance could reach 30 km. When decoy states are employed, 140 km is reached with InGaAs/InP SPDs,

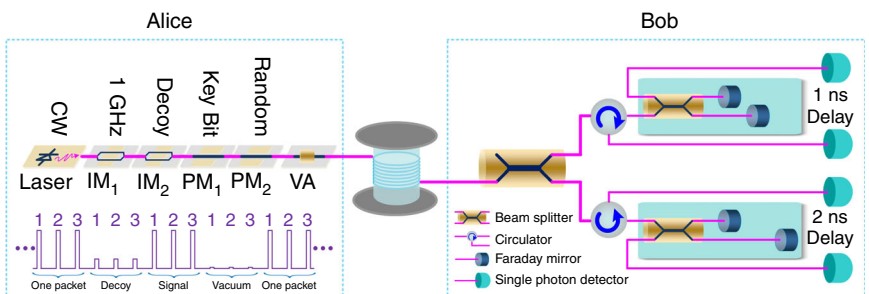

**Fig. 4** Experimental setup to implement the RRDPS protocol with $L = 3$. CW Continuous Wave, IM Intensity Modulator, PM Phase Modulator, VA Variable Attenuator. At Alice's site, a pulse train with a repetition rate of 1 GHz is generated by modulating a 1550.12 nm CW laser using IM$_1$. Every 3 pulses ($L = 3$) are defined as one packet. The intensity of each packet is randomly modulated by IM$_2$ to prepare signal, decoy and vacuum states. PM$_1$ adds phase $-\pi/2$ or $\pi/2$ on each pulse to encode the key bits, and PM$_2$ adds a random global phase on each packet. The VA is used to adjust the average photon-number per pulse. At Bob's site, through a $1 \times 2$ beam-splitter, the incoming signal is randomly coupled into one of two unbalanced Faraday–Michelson interferometers (FMIs) with 1 and 2 ns temporal delays. Each output of the FMIs is led to an SPD. Finally, the detection events are recorded to extract key bits

**Table 3 Experimental results of the $L = 3$ RRDPS system**

| $l$ (km) | $Q_s$ | $E_s$ | $Q_d$ | $E_d$ | $Q_v$ | $R_1$ | $R_2$ |
|---|---|---|---|---|---|---|---|
| 30 | $3.18 \times 10^{-4}$ | 2.32% | - | - | - | $4.20 \times 10^{-6}$ | - |
| 50 | $3.24 \times 10^{-3}$ | 1.76% | $7.52 \times 10^{-4}$ | 1.95% | $1.12 \times 10^{-5}$ | $8.14 \times 10^{-5}$ | $3.60 \times 10^{-4}$ |
| 100 | $3.28 \times 10^{-4}$ | 2.26% | $7.86 \times 10^{-5}$ | 4.01% | $4.50 \times 10^{-6}$ | $4.98 \times 10^{-6}$ | $3.15 \times 10^{-5}$ |
| 140 | $5.52 \times 10^{-5}$ | 4.99% | $1.56 \times 10^{-5}$ | 13.31% | $3.87 \times 10^{-6}$ | - | $1.45 \times 10^{-6}$ |

List of mean yields and error rates of signal ($Q_s$ and $E_s$), decoy ($Q_d$ and $E_d$), and mean yield of "vacuum" ($Q_v$) packets, secret key rates per pulse ($R_1$ and $R_2$) for four lengths of the fiber channel ($l$), where $R_1$ is calculated without using the error rate while $R_2$ is calculated with using the error rate. At 30 km, the decoy states method is not employed, so $Q_s$ and $E_s$ represent the yield per packet and error rate of all key bits, respectively

while the maximum transmission distance of the similar $L = 5$ RRDPS experimental system is less than 50 km with superconducting SPDs[28]. Thus, we have successfully verified the feasibility of RRDPS with the smallest $L = 3$, which is impossible based on the original theory.

## Discussion

We reported on an alternative theory to estimate Eve's information on raw key bits $I_{AE}$. The essence behind our method is that the potential phase randomization can be utilized for the security analysis of RRDPS. The advantage is that $I_{AE}$ can be bounded more tightly than before, especially when $L$ is small. Our results can be used for scenarios both without and with monitoring signal disturbance.

We compared RRDPS with the commonly used BB84 protocol. Although the secret key rate and achievable distance of RRDPS still seem to be inferior to BB84 with decoy states in some typical scenarios, the proposed RRDPS has its particular advantage of post-processing convenience. Moreover, when interferometer misalignment is severe, RRDPS can outperform BB84 significantly. To verify our theory, a proof-of-principle experiment with $L = 3$ is demonstrated here.

There are still several points that should be addressed in the future. In ref. [40], it has been proved that the original bound $I_{AE} \leq h_2(n/(L-1))$ holds with inaccurate phase coding. Our technique depends on the phase randomization, which requires that Alice's phase coding must be 0 or $\pi$ randomly. Therefore, analyzing the relation between phase coding inaccuracy and the upper bound of $I_{AE}$ quantitatively is necessary. Another issue is how to countermeasure the potential attacks due to device imperfections. For example, the blinding attack[41] must be carefully considered in the practical RRDPS systems.

Note added. While preparing the paper, we became aware that similar topics are discussed in other theoretical works[42,43].

## Methods

**Simulation**. We use Wolfram Mathematica 10.3 to run the numerical simulations.

The transmission efficiency of the channel is $\eta = 10^{-\text{loss}/10}$, and loss is just the attenuation (dB) of the channel. Here, we assume that loss stems from the channel while the photon-number-resolving SPDs have 100% efficiency and dark counting rate $d = 10^{-6}$ per pulse. When Bob decides to set the delay value as $r \in \{1, ..., L-1\}$, both of his SPDs will open $L - r$ time-windows to detect the incoming signal. Bob retains only the events in which a single-photon click occurs among these $L - r$ time-windows. The models of the simulations are given below.

We first simulate RRDPS with a single-photon source. Imagining that the delay value is $r$ and both SPDs open one time-window to detect the incoming single-photon, a single click obviously occurs with the probability $(L - r)\eta/L + (1 - (L - r)\eta/L)2d(1 - d)$. The first item represents the probability that the single-photon is not absorbed by the channel or lost due to the noninterfering events in unbalanced Mach–Zehnder interferometers[44,45]. The second item means a dark count occurs, while the single-photon is absorbed by the channel or lost due to the noninterfering events in interferometers. Considering that Bob actually opens $L - r$ time-windows and only retains the case in which there is only one click among these $L - r$ time-windows, the probability that Bob obtains one raw key bit per packet is given by

$$Y_r = (1-d)^{2(L-r)-1}\left(\frac{L-r}{L}\eta + \left(1 - \frac{L-r}{L}\eta\right)2(L-r)d\right). \quad (4)$$

The error rate of the key bit generated by a single-photon packet with delay value $r$ is

$$E_r Y_r = (1-d)^{2(L-r)-1}\left(\frac{L-r}{L}\eta e_{\text{mis}} + \left(1 - \frac{L-r}{L}\eta\right)(L-r)d\right), \quad (5)$$

where $e_{\text{mis}}$ represents the probability that the incoming photon clicks the erroneous SPD due to interferometer misalignment. Accordingly, the mean yield of a single-photon packet is $Y = \sum_{r=1}^{L-1} Y_r/(L-1)$, and its mean error rate is simulated by $EY = \sum_{r=1}^{L-1} E_r Y_r/(L-1)$. The secret key rate per pulse is then given by $RL = Y(1 - h_2(E) - I_{AE}^U)$, where $h_2$ is the information entropy function.

For comparison, we also simulate phase-coding BB84 with a single-photon source here. In a typical phase-coding BB84 system, each encoding state consists of two optical pulses and thus is quite similar to RRDPS with $L = 2$. We then have for phase-coding BB84 its yield per "one pair of pulses" and error rate given by

$$Y = (1-d)\left(\frac{1}{2}\eta + \left(1 - \frac{1}{2}\eta\right)2d\right), \quad (6)$$

and

$$EY = (1-d)\left(\frac{1}{2}\eta e_{\text{mis}} + \left(1 - \frac{1}{2}\eta\right)d\right) \quad (7)$$

respectively, and its secret key rate per pulse is $R = Y(1 - 2h_2(E))/2$.

With a weak coherent source, the method for simulating RRDPS is present here. Assuming the mean photon-number of each pulse emitted by Alice is $\mu$, the mean photon-number per pulse will be attenuated to $\eta\mu$ due to channel loss. When Bob's delay value is $r$, his two SPDs open $L - r$ time-windows to detect $L - r + 1$ weak coherent pulses with mean photon-number $\eta\mu$. Recall the loss due to the noninterfering events in unbalanced Mach–Zehnder interferometers[44,45]; Bob actually attempts to observe the single-photon from $L - r$ weak coherent pulses with mean photon-number $\eta\mu$. For ease of simulation, we assume Bob's SPD can resolve the photon-number perfectly (in practice, one can use the number of double-clicks of threshold SPDs to effectively estimate the number of times Bob receives a multi-photon packet, and then the amount of "tagged" key bits generated by multi-photon receiving events can be upper bounded and further eliminated by privacy amplification[28]). There are then only two possibilities for observing a single-photon click. First, there exists exist only one photon among these $L - r$ weak coherent pulses to click Bob's SPD. The corresponding probability is proportional to $e^{-(L-r)\eta\mu}(L-r)\eta\mu$. Second, there is no photon among these pulses, but Bob's SPD clicks due to dark counting. The corresponding probability is then proportional to $e^{-(L-r)\eta\mu}$. Summing over the two possibilities and recalling that Bob retains only the case in which there is only one click among these $L - r$ time-windows, we have

$$Q_r = (1-d)^{2(L-r)-1}e^{-(L-r)\eta\mu}((L-r)\eta\mu + 2(L-r)d), \quad (8)$$

and the overall counting rate $Q = \sum_{r=1}^{L-1} Q_r/(L-1)$. The error rate $E$ can be simulated by

$$EQ = \sum_{r=1}^{L-1}\frac{1}{L-1}(1-d)^{2(L-r)-1}e^{-(L-r)\eta\mu}((L-r)\eta\mu e_{\text{mis}} + (L-r)d). \quad (9)$$

In the case without monitoring signal disturbance and decoy states, the secret key rate $R$ per pulse is given by

$$RL = Q(1 - h_2(E)) - e_{\text{src}} - (Q - e_{\text{src}})I_{AE}^U, \quad (10)$$

where $e_{\text{src}} = 1 - \sum_{n=0}^{\nu_{\text{th}}} e^{-L\mu}(L\mu)^n/n!$ is the probability that the photon-number of a packet is greater than $\nu_{\text{th}}$. In our method, $I_{AE}^U$ is calculated by Eq. (1) setting the photon-number $N = \nu_{\text{th}}$ and ignoring constraint Eqs. (2) and (3). In the original method, $I_{AE}^U = h_2(\nu_{\text{th}}/(L-1))$. $\mu$ and $\nu_{\text{th}}$ should be optimized to maximize $R$.

For comparison, we also simulate phase-coding BB84 with a weak coherent source here. Note we assume that infinite decoy states are employed in this case. Similar to the method of obtaining Eqs. (10) and (11) in ref. [39], we have for phase-

coding BB84 its yield per "one pair of pulses" and error rate given by

$$Y_n = \left(1 - \frac{1}{2}\eta\right)^{n-1}(1-d)\left(\frac{1}{2}n\eta + \left(1 - \frac{1}{2}\eta\right)2d\right), \quad (11)$$

and

$$E_n Y_n = \left(1 - \frac{1}{2}\eta\right)^{n-1}(1-d)\left(\frac{1}{2}n\eta e_{\text{mis}} + \left(1 - \frac{1}{2}\eta\right)d\right), \quad (12)$$

respectively, where $n$ is the photon-number of the encoding state. Summing over all possible photon-numbers $n$, its mean yield and error rate are given by $Q = \sum_{n\geq 0} e^{-2\mu}(2\mu)^n Y_n/n!$ and $EQ = \sum_{n\geq 0} e^{-2\mu}(2\mu)^n E_n Y_n/n!$, respectively, where $\mu$ is the mean photon-number per pulse. Its secret key rate is

$$2R = -Qh_2(E) + e^{-2\mu}2\mu Y_1(1 - h_2(E_1)). \quad (13)$$

In an experiment of RRDPS with $L = 65$ given in ref. [29], there is a set of experimental observations: the mean photon-number $s = 0.037$ per pulse, yield $Q_s = 8.435 \times 10^{-4}$ per packet and error rate $E = 0.058$. By setting $v_{\text{th}} = 10$, the secret key rate is $R_1 = (Q_s(1 - 1.1h_2(E)) - e_{\text{src}} - (Q_s - e_{\text{src}})h_2(v_{\text{th}}/64))/L = 5 \times 10^{-8}$. With the same parameters and finding $I_{AE}^U = 0.513$ for 10-photon, $R_2 = (Q_s(1 - 1.1h_2(E)) - e_{\text{src}} - (Q_s - e_{\text{src}})I_{AE}^U)/L = 1.44 \times 10^{-6}$.

**Key rate for the experiment**. Here, we describe the methods for obtaining the secret key rates in Table 3.

At 30 km, the system was running without monitoring and decoy states. Its secret key rate $R$ is then calculated by $RL = Q_s(1 - h_2(E)) - e_{\text{src}} - (Q_s - e_{\text{src}})I_{AE}^U$, where $e_{\text{src}} = 1 - e^{-L\mu} - L\mu e^{-L\mu}$, $\mu = 0.005$, and $I_{AE}^U$ is calculated by formula (1).

At the other distances, decoy states were employed. The photon-numbers per pulse of signal, decoy, and "vacuum" packets are given the values $s = 0.13$, $d = 0.03$, and $v = 0.0003$, respectively. In the experiment, we directly observe the yields $Q_s$, $Q_d$ and $Q_v$ for signal, decoy and "vacuum" packets, respectively. The error rates $E_s$ ($E_d$) for key bits generated from signal (decoy) packets are also observed experimentally. Referring to ref. [39], we can estimate the yield $Y_1$ for single-photon packets and the error rate $E_1$ for key bits generated from single-photon packets by the following:

$$Y_0 = \max\left\{\frac{LdQ_v e^{Lv} - LvQ_d e^{Ld}}{Ld - Lv}, 0\right\} \quad (14)$$

$$Y_1 = \frac{Ls}{LsLd - LsLv - (Ld)^2 + (Lv)^2}\left(Q_d e^d - Q_v e^v - \frac{(Ld)^2 - (Lv)^2}{(Ls)^2}(Q_s e^s - Y_0)\right), \quad (15)$$

$$E_1 = \frac{E_s Q_s e^{Ls} - E_d Q_d e^{Ld}}{(Ls - Ld)Y_1}. \quad (16)$$

As a proof-of-principle experiment, the secret key rates $R_1$ and $R_2$ in Table 3 are not obtained by actually performing post-processing steps. Instead, they are calculated by $R = (Lse^{-Ls}Y_1(1 - I_{AE}^U) - Q_s h_2(E_s))/L$. Here, to obtain $R_1$, we calculate $I_{AE}^U$ with Eq. (1), while ignoring constraint Eqs. (2) and (3). To $R_2$, this constraint with $E = E_1$ is used.

**Code availability**. Source codes of the plots are available from the corresponding authors on request.

**Data availability**. The data that support the findings of this study are available from the corresponding authors on request.

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

## Acknowledgements

We would like to thank Prof. Xiongfeng Ma, Dr Xiao Yuan, and Dr Zhu Cao for helpful discussions. This work has been supported by the National Key Research And Development Program of China (Grant Nos. 2016YFA0302600 and 2016YFA0301702), the National Natural Science Foundation of China (Grant Nos. 61775207, 61475148, 61627820, 61622506, 61575183, and 61675189), the "Strategic Priority Research Program (B)" of the Chinese Academy of Sciences (Grant No. XDB01030100).

## Author contributions

Z.-Q.Y., S.W., W.C., G.-C.G., and Z.-F.H. conceived the basic idea of the security proof and designed the experiments. Z.-Q.Y. finished the details of the security proof. S.W. designed the variable-delay interferometer. W.C. designed the control and detection parts of the system. Z.-Q.Y., Y.-G.H., and R.W. designed the simulations. Z.-Q.Y. and S.W. wrote the paper.
