## [Peer Review File · Nature Communications]

Reviewers' comments:

Reviewer #1 (Remarks to the Author):

In this manuscript, the authors propose a new method for analyzing the security of the RRDPS protocol. The analysis is applicable only to the case where Alice emits a single photon in the block of L pulses. Compared to the previously known formula for the same situation, the required amount I_{AE} of privacy amplification is smaller. Further, the functional dependence of I_{AE} on the observed bit error rate E is calculated, which allows the use of $I_{AE}(E)$ instead of the worst of $I_{AE}(E)$ over E . They also report experimental demonstration of the RRDPS protocol with $L=3$ augmented with decoy-state technique, and showed that a distance of 140 km is feasible.

Among the two major improvements in the proposed theory, the E dependence of I_{AE} is not so original. For one thing, the main feature of the RRDPS protocol is the fact that the formula can use I_{AE} that is independent of E , and hence its E dependence was not the focus of interest. Once we are interested in the E dependence, the derivation seems to be not so difficult. In fact, there is another paper (arXiv:1701.08509) which calculates $I_{AE}(E)$ in a different analytical method.

On the other hand, the improvement of I_{AE} (the worst value over E) is quite interesting, because it suggests that the current formula for the RRDPS protocol may have a room for significant improvement.

I then asked myself whether the importance of the improvement that was already achieved in this work is sufficient for publication in Nature Communications. The answer is negative, mainly due to the limitation of the theory to the single-photon emission. Let us compare the original RRDPS protocol to the decoy-state BB84 protocol. In terms of the asymptotic key rate at a low bit error rate, the decoy-state BB84 protocol is the better. Hence the advantage of the original RRDPS protocol lies in the finite-key effects and in the high tolerance on bit error rates. Now, due to its limitation to the single-photon emission, the actual implementation of the present work must be accompanied by the decoy-state technique. This may kill the advantage in the finite-key effects, as the authors stated in the conclusion. As for the other advantage, I should point out that high bit-error tolerance of single-photon QKD schemes has long been discussed in terms of qudit- or multidimensional-QKD schemes, which is totally ignored in the current manuscript. For example, there is a paper by Chau (Phys. Rev. A 92, 062324) claiming that it can tolerate up to 50 % bit-error rate.

To summarize, although the manuscript proposes an interesting method which will deepen the understanding of the RRDPS protocol and hence it will appeal to the audience who are already interested in the protocol, I feel it falls short of capturing the interest of general readers of the Nature Communications.

Reviewer #2 (Remarks to the Author):

The work offers an improved security proof for the round-robin QKD protocol proposed in Nature 509, 475 (2014). The essence of the work relies on improving the upper bound of the leaked information to an eavesdropper, I_{AE} , by accounting for the QBER in the link. As a result, they are able to get positive key rates at long distances even if a short train of L pulses, with L being around 3 to 5, is used. The authors claim that this would make the round-robin protocol more practical. Despite of some typos and inconsistency in the notation, the bulk of the analysis seems to be correct and sensible. I have, however, concerns about the importance of this work. Here are the key points to support my assessment.

1) The key feature of the original round-robin QKD protocol is its independence from the parameter estimation that specifies how much privacy amplification is needed. In that scheme, I_{AE} is upper bounded by $h(1/(L-1))$, with h being the binary Shannon entropy function, in the ideal single-photon scenario. In the original BB84 protocol, I_{AE} is bounded by $h(e_p)$, where e_p is the phase error and needs to be estimated. The advantage of the round-robin protocol is that some post-processing would become easier if we do not need to estimate e_p . It would also offer positive key rates, for large L , even if e_p is greater than 11%. The key rate obtained from the round-robin protocol would, however, be lower than that of original BB84 protocol if $e_p < 1/(L-1)$. That, and other factors, imply that to see some rate advantage over the BB84 protocol, L should be roughly greater than 16.

2) In this work, the claim is that using the improved security analysis, we can now even use the round-robin QKD protocol at $L=3$. This is correct, but the key question is that what advantage it would have over already established protocols like decoy-state BB84. There seems to be no distance or rate advantage as even the improved key rate that can be obtained from the given analysis is lower than that of the decoy-state protocol at $L=3-5$. Plus, the authors' proposed analysis requires estimation of parameters like e_1 and Q_1 in the decoy-state protocol, hence does not offer any post-processing advantage either.

My conclusion from points (1) and (2) is that for the round-robin QKD protocol to be of relevance it should be operated at rather large L (roughly >10).

3) The authors claim that at large L , there would be little advantage in using their result as $1/(L-1)$ is already a small number. That means that the additional post-processing required by the proposed analysis does not buy us much, and it is not really relevant.

My conclusion, based on the above points and the presented results, is that the proposed security bound offers little, if not none, practical advantage to the existing stack of QKD protocols in terms of their rate and reach. There might be a region that at a decent value of L , we can use the proposed analysis to beat decoy-state BB84 while getting some substantial advantage by using the proposed improved bound for I_{AE} , but this has not been demonstrated in the paper and requires further numerical investigation. I am therefore not convinced that the paper suits publication in Nature Communications. On the technical side, the paper perhaps merits publication in some other journals, but that also requires improvement in the presentation aspects of the paper.

Reviewer #3 (Remarks to the Author):

Referee report for manuscript NCOMMS-17-03213. This manuscript has 2 main points, one that they have improved on the security proof for Round-robin-differential-phase-shift QKD to include a dependency on the error rate. The second is they did an experimental demonstration of RRDPS QKD in a distance regime that was previously undemonstrated. While the results are new, they are so poorly presented that I cannot recommend publishing this in Nature Communication or anywhere else. I will go through just some of the key reasons why below.

Firstly, the argument is made in the introduction that RRDPS is a desirable protocol to look at as the previously known bounds on an eavesdropper's knowledge of the key was not dependent on the error rate. They proceed to show that by modifying the security proof to include the error rate, they can demonstrate RRDPS at longer distances and with less components than previous. This is fine, but seems to then remove the aspect of protocol that made it unique. The introduction needs to make a stronger case/have more discussion about different QKD protocols and their limitations. Along those lines, the paper in general had an extremely low number of citations, with a quite a few self

citations. A paper's results are only useful if the authors can put it in context of the broader field, and not just their own work.

Secondly, I have no way to evaluate the simulation section of the paper. No details of how the simulations were done were given (the two citations listed also had no details about any simulations). Ideally their source code is provided as supplementary, or they must explain why the code cannot be made available and describe it in sufficient detail with pseudo code such that I can actually evaluate these results. Providing source code is a policy of Nature outlined here:

<<http://www.nature.com/authors/policies/availability.html#code> > and

<<http://www.nature.com/news/announcement-trans>> Similarly, there were no citations to the software that they used.

Lastly, with the experimental section, I cannot really make heads or tails of the results table. The main point of the experimental tests at these longer distances and "simplest" setup (which no definition is given for or argument made that it is the simplest) is that you still have a positive key rate. I think there is a key rate number quoted in the table, but I have no idea if this is a projected key rate or they actually did full classical post-processing of the data and extracted keys. One key additional piece of the results for readers is the actual data. As described in the links above, Nature has standards for open data and code and I think especially for this work it is critical to include both. Quoting from the second link above: "The guidelines consist of eight standards — citation standards, data transparency, analytic methods (code) transparency, research materials transparency, design and analysis transparency, pre-registration of studies, preregistration of analysis plans, and replication — with three levels of increasing rigour."

Again, because there is no detail or data provided on one of the most critical claims of the paper, I cannot evaluate the validity of the results.

In summary, this work may be interesting and novel, I have no way of evaluating it from the provided manuscript.

To Reviewer #1

We thank Reviewer #1 for the careful reading of our paper and objective comments. A main contribution of the revised version is extending our theory to general N-photon cases. Thus, decoy states are not indispensable any more. Our responses to the comments are given in blue color as follows.

1. In this manuscript, the authors propose a new method for analyzing the security of the RRDPS protocol. The analysis is applicable only to the case where Alice emits a single photon in the block of L pulses. Compared to the previously known formula for the same situation, the required amount I_{AE} of privacy amplification is smaller. Further, the functional dependence of I_{AE} on the observed bit error rate E is calculated, which allows the use of $I_{AE}(E)$ instead of the worst of $I_{AE}(E)$ over E. They also report experimental demonstration of the RRDPS protocol with L=3 augmented with decoy-state technique, and showed that a distance of 140 km is feasible.

Among the two major improvements in the proposed theory, the E dependence of I_{AE} is not so original. For one thing, the main feature of the RRDPS protocol is the fact that the formula can use I_{AE} that is independent of E, and hence its E dependence was not the focus of interest. Once we are interested in the E dependence, the derivation seems to be not so difficult. In fact, there is another paper (arXiv:1701.08509) which calculates $I_{AE}(E)$ in a different analytical method.

On the other hand, the improvement of I_{AE} (the worst value over E) is quite interesting, because it suggests that the current formula for the RRDPS protocol may have a room for significant improvement.

Response: Actually, while uploading the paper to arXiv, we are aware that similar

topics are discussed in the two theoretical works “arXiv:1701.08509” as mentioned by the reviewer #1 and “arXiv:1702.00162”. I have cited them in the revised version. However, our method is completely different from theirs. We directly construct Eve’s optimal attack, but the other two papers are based on the calculation of phase error rate. Most importantly, our bound of I_{AE} is tighter than theirs, which implies that with our method, higher key rate and longer achievable distance are both expected.

2. I then asked myself whether the importance of the improvement that was already achieved in this work is sufficient for publication in Nature Communications. The answer is negative, mainly due to the limitation of the theory to the single-photon emission. Let us compare the original RRDPS protocol to the decoy-state BB84 protocol. In terms of the asymptotic key rate at a low bit error rate, the decoy-state BB84 protocol is the better. Hence the advantage of the original RRDPS protocol lies in the finite-key effects and in the high tolerance on bit error rates. Now, due to its limitation to the single-photon emission, the actual implementation of the present work must be accompanied by the decoy-state technique. This may kill the advantage in the finite-key effects, as the authors stated in the conclusion. As for the other advantage, I should point out that high bit-error tolerance of single-photon QKD schemes has long been discussed in terms of qudit- or multidimensional-QKD schemes, which is totally ignored in the current manuscript. For example, there is a paper by Chau (Phys. Rev. A 92, 062324) claiming that it can tolerate up to 50 % bit-error rate.

To summarize, although the manuscript proposes an interesting method which will deepen the understanding of the RRDPS protocol and hence it will appeal to the audience who are already interested in the protocol, I feel it falls short of capturing the interest of general readers of the Nature Communications.

Response: We agree that the main drawback of our work is its limitation to the single-photon emission, which will kill some important advantages of RRDPS protocol. Through thoughtful analyses and calculations, we have extended our theory to general N-photon case. A very detailed security proof under N-photon case is presented in the supplementary file, while the main results are generalized as a theorem given in the section II.A of the revised manuscript. This theorem and its corollary give a very concise bound of I_{AE} which can be used in scenarios both without and with monitoring signal disturbance. Based on this theorem and numerical simulation, we show that with the help of our method, the performance of RRDPS can be improved dramatically while its main advantages such as no need of decoy states, high bit-error tolerance, easier post-processing are all still available.

Besides, we apologize for the ignorance of multi-dimensional QKD and recently proposed Chau15 protocol. We have cited these valuable works in the revised version. We totally agree that Chau15 is a striking protocol due to its distinct tolerance for high error rate.

In a work, we think the main disadvantage of our work, as pointed by the reviewer #1, has been solved thoroughly in the revised version.

To Reviewer #2

We thank Reviewer #2 for the careful reading of our paper and helpful comments. A main contribution of the revised version is extending our theory to general N-photon cases. Thus, decoy states are not indispensable any more. Our responses to the comments are given in blue color as follows.

The work offers an improved security proof for the round-robin QKD protocol proposed in Nature 509, 475 (2014). The essence of the work relies on improving the upper bound of the leaked information to an eavesdropper, I_{AE} , by accounting for the QBER in the link. As a result, they are able to get positive key rates at long distances even if a short

train of L pulses, with L being around 3 to 5, is used. The authors claim that this would make the round-robin protocol more practical. Despite of some typos and inconsistency in the notation, the bulk of the analysis seems to be correct and sensible. I have, however, concerns about the importance of this work. Here are the key points to support my assessment.

1) The key feature of the original round-robin QKD protocol is its independence from the parameter estimation that specifies how much privacy amplification is needed. In that scheme, I_{AE} is upper bounded by $h(1/(L-1))$, with h being the binary Shannon entropy function, in the ideal single-photon scenario. In the original BB84 protocol, I_{AE} is bounded by $h(e_p)$, where e_p is the phase error and needs to be estimated. The advantage of the round-robin protocol is that some post-processing would become easier if we do not need to estimate e_p . It would also offer positive key rates, for large L , even if e_p is greater than 11%. The key rate obtained from the round-robin protocol would, however, be lower than that of original BB84 protocol if $e_p < 1/(L-1)$. That, and other factors, imply that to see some rate advantage over the BB84 protocol, L should be roughly greater than 16.

Response: We agree that based on the original paper of RRDPS, an L greater than 16 is indispensable to see advantages of RRDPS. In my opinion, it is important to decrease the value of L while keeping its secret key rate and achievable distance. A main contribution of our work is the improvement on key rate and achievable distance compared to the original RRDPS with the same L . In other words, based on our work, RRDPS with a relatively smaller L can be better than the original RRDPS with a larger L . For instance, the simulation in the revised version shows that with the help of our theory, the RRDPS with $L=16$ has advantages on key rate and achievable distance compared to the original one with $L=32$.

2) In this work, the claim is that using the improved security analysis, we can now even use the round-robin QKD protocol at $L=3$. This is correct, but the key question is that what advantage it would have over already established protocols like decoy-state BB84. There seems to be no distance or rate advantage as even the improved key rate that can be obtained from the given analysis is lower than that of the decoy-state protocol at $L=3-5$. Plus, the authors' proposed analysis requires estimation of parameters like e_1 and Q_1 in the decoy-state protocol, hence does not offer any post-processing advantage either.

My conclusion from points (1) and (2) is that for the round-robin QKD protocol to be of relevance it should be operated at rather large L (roughly >10).

Response: We agree there will be no advantage over commonly used BB84, if we employ a RRDPS implementation with $L=3$. In the revised version, we made detailed simulation and comparison for the cases $L>10$. The proof-of principle experiment with $L=3$ is retained to just verify that our theory is valid for the smallest L . And we remove any misleading statements that larger L is not necessary.

What we stress here is that we have extended our theory to general N -photon case in the revised version, which implies that our method can be applicable in the scenarios without monitoring signal disturbance and decoy states. Hence the post-processing advantage of RRDPS remains valid now. Through numerical simulation, we show that the key rate and distance can be both improved dramatically compared to original RRDPS. Besides, when optical interference is worse, the RRDPS also gets substantial advantages over the BB84.

3) The authors claim that at large L , there would be little advantage in using their result as $1/(L-1)$ is already a small number. That means that the additional post-processing required by the proposed analysis does not buy us much, and it is not really relevant.

Response: We apologize for this irrelevant claim. This claim has been removed in the revised manuscript.

My conclusion, based on the above points and the presented results, is that the proposed security bound offers little, if not none, practical advantage to the existing stack of QKD protocols in terms of their rate and reach. There might be a region that at a decent value of L , we can use the proposed analysis to beat decoy-state BB84 while getting some substantial advantage by using the proposed improved bound for I_{AE} , but this has not been demonstrated in the paper and requires further numerical investigation. I am therefore not convinced that the paper suits publication in Nature Communications. On the technical side, the paper perhaps merits publication in some other journals, but that also requires improvement in the presentation aspects of the paper.

Response: We agree that the presented results in the last version offer little advantage to the existing protocols. In the revised version, our method has been greatly improved and can be used for general N photon cases. Accordingly, we made further numerical investigations as required.

Compared with the original DPS protocol: Under the condition that decoy states and monitoring signal disturbance are totally bypassed in RRDPS, we compare the original and proposed RRDPS in terms of key rate and achievable distance. In Fig.1 (with optical misalignment parameter $e_{\text{mis}}=0.015$) and Fig.2 (with $e_{\text{mis}}=0.15$) of the revised manuscript, the key rates versus channel losses for RRDPS with three typical values 16, 32, and 64 of L are presented. We see that the proposed methods improve the performance of the original RRDPS dramatically.

Compared with the BB84 + infinite decoy states protocol: When optical misalignment parameter $e_{\text{mis}}=0.015$, we find that the BB84 still overwhelms the RRDPS in terms of key rate and achievable distance, as illustrated in Fig. 1 of the

manuscript. However, the latter one works without monitoring signal disturbance and decoy states, which is very meaningful for the post-processing of QKD systems and alleviating the finite key-size effects as mentioned by the reviewer #1. Moreover, we also demonstrate that the RRDPS with our methods is still able to distribute secret key bits in practice when $e_{\text{mis}}=0.15$, while BB84 cannot generate any secret key bit in this case. In a word, the proposed protocol outperforms the BB84 significantly when $e_{\text{mis}}=0.15$. Hence, when optical interference is worse, the RRDPS gets substantial advantages over the BB84. The situations that optical interference is worse must be considered in many important applications. For examples, the environmental disturbance or electrical noises in high speed QKD systems will lower the visibility of interference. As far as I know, there is no other protocol can tolerate such a high error rate while without monitoring signal disturbance is required.

In summary, through further numerical simulations, we demonstrate that the proposed method improves the rate and reach of the original RRDPS substantially while its advantages of post-processing and no need for decoy states are retained. Besides, we also show under high optical misalignment situations, the proposed protocol beats other QKD protocols, such as BB84.

To Reviewer #3

We thank Reviewer #3 for the careful reading of our paper and valuable comments. A main contribution of the revised version is extending our theory to general N-photon cases. Thus, decoy states are not indispensable any more. Our responses to the comments are given in blue color as follows.

Referee report for manuscript NCOMMS-17-03213. This manuscript has 2 main points, one that they have improved on the security proof for Round-robin-differential-phase-shift QKD to include a dependency on the error rate. The second is they did an experimental demonstration of RRDPS QKD in a

distance regime that was previously undemonstrated. While the results are new, they are so poorly presented that I cannot recommend publishing this in Nature Communication or anywhere else. I will go through just some of the key reasons why below.

Firstly, the argument is made in the introduction that RRDPDS is a desirable protocol to look at as the previously known bounds on an eavesdropper's knowledge of the key was not dependent on the error rate. They proceed to show that by modifying the security proof to include the error rate, they can demonstrate RRDPDS at longer distances and with less components than previous. This is fine, but seems to then remove the aspect of protocol that made it unique. The introduction needs to make a stronger case/have more discussion about different QKD protocols and their limitations. Along those lines, the paper in general had an extremely low number of citations, with a quite a few self citations. A paper's results are only useful if the authors can put it in context of the broader field, and not just their own work.

Response: We apologize for neglecting other QKD protocols and lack of relevant citations. Now, introductions on many other QKD protocols, e.g. DI, MDI, COW, DPS, are added. Besides, a brief introduction on the experimental progresses is also included. The reviewer #3 said “This is fine, but seems to then remove the aspect of protocol that made it unique.” We stress that in the revised version our theory can tackle the general N-photon case, which means that our method can be effectively used without decoy states and monitoring signal disturbance, and thus the unique of the protocol is retained now.

However, our proof-of-principle experiment still runs with $L=3$ and decoy states, since this experiment aims to verify that L can be lowered to such a small value, which is not permitted in the original RRDPDS protocol. Indeed, to obtain distance or rate advantage over BB84 and bypass monitoring signal disturbance and decoy states, a larger L is necessary.

Secondly, I have no way to evaluate the simulation section of the paper. No details of how the simulations were done were given (the two citations listed also had no details about any simulations). Ideally their source code is provided as supplementary, or they must explain why the code cannot be made available and describe it in sufficient detail with pseudo code such that I can actually evaluate these results. Providing source code is a policy of Nature outlined here: <http://www.nature.com/authors/policies/availability.html#code> and <http://www.nature.com/news/announcement-trans>. Similarly, there were no citations to the software that they used.

Response: We apologize for missing details of the simulation. The details of the simulation are presented in methods section now. Besides, we are willing to upload the Wolfram Mathematica source code, by which the reviewer #3 can evaluate it. The “calculateIAE.nb” file contains the functions to calculate I_{AE} with different L and photon-number N. This file must be run before running other files. The “noerrorate.nb” and “witherrorate.nb” files are used to simulate key rate versus channel loss without and with monitoring signal disturbance respectively. Furthermore, “noerrorate.opj” and “witherrorate.opj” files contain the output data of the simulations. The file “forL=65.nb” is used to calculate key rate based on the already existing experiment with L=65.

Lastly, with the experimental section, I cannot really make heads or tails of the results table. The main point of the experimental tests at these longer distances and "simplest" setup (which no definition is given for or argument made that it is the simplest) is that you still have a positive key rate. I think there is a key rate number quoted in the table, but I have no idea if this is a projected key rate or they actually did full classical post-processing of the data and extracted keys. One key additional piece of the results for readers is the actual data. As described in the links above,

Nature has standards for open data and code and I think especially for this work it is critical to include both. Quoting from the second link above: "The guidelines consist of eight standards — citation standards, data transparency, analytic methods (code) transparency, research materials transparency, design and analysis transparency, pre-registration of studies, preregistration of analysis plans, and replication — with three levels of increasing rigour." Again, because there is no detail or data provided on one of the most critical claims of the paper, I cannot evaluate the validity of the results.

In summary, this work may be interesting and novel, I have no way of evaluating it from the provided manuscript.

Response: We apologize for missing some important points of the experiment. The error rates of key bits and yields per packet are directly obtained experimentally. Then we use standard decoy states formulae to calculate the yield and error rate for single photon packet. Finally the secret key rates R_1 and R_2 are calculated according to our theory. As a proof-of-principle experiment, the secret key rates R_1 and R_2 in Tab.II are calculated according to our theory, but not obtained by actually performing post-processing steps. In my opinion, the essential of our work is the novel theory and its expected improvements of RRDPS, which have nothing to do with if we actually perform the post-processing.

The details of the experiment are presented in methods section of the revised version. Besides, we upload the "forexperiment.nb" which is used to calculate the key rate based on our experimental data. I hope the reviewer #3 is able to evaluate our results now.

Reviewers' comments:

Reviewer #1 (Remarks to the Author):

My main concern in the previous report, the limitation of the theory to the single-photon emission, has been lifted in the revised manuscript. This makes me believe that the manuscript is suitable for publication in Nature Communications.

Reviewer #2 (Remarks to the Author):

Reading the revised version of the paper, I think the paper key contribution is in tightening the upper bound on I_{AE} . They consider two cases with and without error monitoring. The case of without monitoring is the more interesting case for me as they show, with their improved security proof, it is now possible to obtain secret keys even at a low value of $L=3$, which previously was not possible. They also show that they can tolerate higher error rates than the originally proposed RRDPS protocol.

I am less impressed with their results of the "with monitoring" case. In terms of post-processing convenience, monitoring would require the same steps as in BB84 with decoy states without necessarily offering any practical advantages in terms of rate and reach. The curves in Fig. 3 all perform below the decoy-state QKD. Even in Fig. 2, where misalignment has got an unreasonably high value of 0.15, we need $L \geq 16$ to get some advantage, which is against the objective that we can use this system at low values of L . The case of $L=3$ tolerates less error rate than a typical BB84 at 11%.

Based on the above observations, my understanding is that, while the improved security proof enables us to operate at low values of L , such low- L systems are still inferior to established QKD techniques such as BB84 with decoy states. I am sure the results are worth publishing somewhere, but whether the impact is sufficiently high to secure a spot in Nature Commun., it's not a clear call for me.

I think the paper, whether or not it finds a place in Nature Commun., needs to be rewritten to better reflect the key contribution. There needs to be some clarifications and some restructuring and rewording for the paper to better present its results. I think historically the authors have first found the results with monitoring, and that is why they insist in including it. Whereas, the key results for me is in column three of Table I (Eq. (1) without E). Other minor suggestions:

- The bound on I_{AE} is different from I_{AE} itself. In many places the difference has been ignored and I_{AE} refers to its upper bound without being explicit.

- I think on page 2, it should be de Finetti theorem (not defitti)

- The notation and explanation on page 3 for the N-photon case is not clear.

- On Pg 4, "tolerance of E" needs to be clearly defined.

- e_{mis} needs to be clearly defined in such time/phase encoding techniques.

- I think the pulse width is assumed constant for different values of L. This should be clarified in the text.

- The choice of $e_{mis} = 0.15$ mus be justified by giving a practical scenario in which this is expected. Alternatively, if the authors want to show the advantage of their proof, they should find more realistic examples where by a choice of $L < 10$, they get some advantage out of their system.

- For me Fig. 3 does not have much to offer. We use a complicated system with $L=16$ plus monitoring, and we still do not outperform BB84+decoy.

- Pg 6, what kind of optimization VA does? is it for different distances?

Reviewer #3 (Remarks to the Author):

Referee report for manuscript NCOMMS-17-03213. I will just be responding to the changes made in my first report.

First point: Put results/Choice of protocol in context for the field.

The authors have done the barest minimum to address this point, in that I feel they just laundry list a bunch of other protocols. They make it clear that RRDPs does not need to measure the error rate, but do not say why that is helpful. Is it hard to measure this rate? I guess as a reader, I would still be unconvinced as to why I should use RRDPs.

Second point: The additional details in the methods section are greatly appreciated, however there should be more citations for where the formulas come from in this section. I am very glad the authors were willing to upload their code. Unfortunately, I do not have a licence to Mathematica so I am unable to actually run or evaluate the code. That said, I would commend the authors for sharing the code because this is critical to good science and the review process.

Third point: The same response to point 2 applies here. I cannot run the code, as I do not have a licence to this software. I understand that the authors do not implement their own post-processing and this is fine, as long as this is stated and the formulas used to estimate the secret key rate are stated.

I unfortunately cannot make a recommendation on this paper, because I cannot actually evaluate the code, which supports large sections of the paper.

We thank the Reviewers for the helpful comments. The corresponding replies are given in blue color as follows. The modifications of the manuscript have been highlighted in yellow.

To Reviewer #1

My main concern in the previous report, the limitation of the theory to the single-photon emission, has been lifted in the revised manuscript. This makes me believe that the manuscript is suitable for publication in Nature Communications.

Response: We thank Reviewer #1 for his/her recommendation.

To Reviewer #2

We thank Reviewer #2 for the careful reading of our paper and helpful comments. Accordingly, major revisions of the manuscript, especially the results section, have been made to focus more on the 'without-monitoring' case.

Reading the revised version of the paper, I think the paper key contribution is in tightening the upper bound on I_{AE} . They consider two cases with and without error monitoring. The case of without monitoring is the more interesting case for me as they show, with their improved security proof, it is now possible to obtain secret keys even at a low value of $L=3$, which previously was not possible. They also show that they can tolerate higher error rates than the originally proposed RRDPs protocol.

Response: Yes, the essential point of this work is to estimate the upper bound on I_{AE} more tightly. Consequently, the positive key rate can be achieved even when $L=3$. And RRDPs can tolerate higher error rates than before.

Hence, with the help of our theory, both the secret key rate and the achievable

distance of RRDPS protocol can be improved.

I am less impressed with their results of the "with monitoring" case. In term of post-processing convenience, monitoring would require the same steps as in BB84 with decoy states without necessarily offering any practical advantages in terms of rate and reach. The curves in Fig. 3 all perform below the decoy-state QKD.

Response: We admit that monitoring signal disturbance causes that RRDPS loses its particular post-processing convenience, while no advantages over BB84+decoy states in terms of rate and reach are observed. Thus, the curves of RRDPS with monitoring signal disturbance are removed in the revised version.

Theoretically speaking, it is an outstanding character that our formula (1) and (2) can calculate I_{AE} both without and with monitoring signal disturbance, while other theoretical works on RRDPS cannot estimate I_{AE} in this way. Hence, our result may be a unique reference for the theoretical research of RRDPS.

Based on above considerations, we retain the theoretical results on monitoring signal disturbance, i.e. formula (2), while most simulations with monitoring signal disturbance are removed in the revised version.

Even in Fig. 2, where misalignment has got an unreasonably high value of 0.15, we need $L \geq 16$ to get some advantage, which is against the objective that we can use this system at low values of L .

Response: In the new Fig.2, a curve for $L=8$ is added, which shows that we can use RRDPS system at low values of L . More importantly, from Fig.2 we can see that for an RRDPS system with typical values $L = 8, 16, 32$, our formula (1) does improve its performance in terms of secret key rate and achievable distance.

The reasons for why we should consider misalignment with such high value will be elaborated later.

The case of $L=3$ tolerates less error rate than a typical BB84 at 11%.

Response: Yes, RRDPS with $L=3$ is inferior to BB84. However, with the smallest value of L , the case of $L=3$ should be retained here, in my opinion. The reasons are as follows.

The $L=3$ case is meaningful when we compare our results with the original RRDPS, which cannot generate any secret key bit at $L=3$.

Besides, the proposed RRPDS with $L=3$ can run without monitoring, while it is impossible for BB84. To justify this point, we add an additional experiment for $L=3$ without monitoring signal disturbance and decoy states. It is demonstrated that RRDPS with $L=3$ can generate key bit at the distance of 30 km, while both decoy states and monitoring are still turned off. This additional experiment can strengthen the significance of $L=3$ case and help the manuscript focus more on ‘without monitoring signal disturbance’ case.

Based on the above observations, my understanding is that, while the improved security proof enables us to operate at low values of L , such low- L systems are still inferior to established QKD techniques such as BB84 with decoy states. I am sure the results are worth publishing somewhere, but whether the impact is sufficiently high to secure a spot in Nature Commun., it's not a clear call for me.

Response: The significance of our work can be summarized with four points.

1) In theory, we develop a method to bound information leakage of RRDPS quite tightly and differently. As far as we know, our bound of I_{AE} given by formula (1) without monitoring signal disturbance is tighter than any other published results. Interestingly, the dependence between Eve's information on key bits and signal disturbance is also clarified, which is significant for the understanding of RRDPS. Thus, this result may be most outstanding one among the works on the security proof of RRDPS.

2) With the help of our theory, we prove that the performance of RRDPS can be improved dramatically, while its advantage of without monitoring signal disturbance

can be still available. And its demand on large L values is greatly alleviated, by which RRDPS can become more practical. We believe that the demonstration of this method can show a new way to improve monitoring-free QKD protocols.

3) Although the rate and reach of RRDPS seem to be still inferior to BB84+decoy states in some typical scenarios, the proposed RRDPS has its particular advantage of post-processing convenience, especially for practical systems running in the volatile environments when taking the finite-sized-key effects into account, which is impressive for the research of QKD.

4) We should note that in the scenarios that optical misalignment is severe, e.g. $e_{\text{mis}} > 10\%$, our protocol does outperform BB84+decoy states. The reasons for why we should consider high e_{mis} cases will be elaborated later.

Because of the highlights in both theoretical and experimental QKD, we believe that our work will be an extraordinary reference for researchers of quantum communications, and will be of interest to a wide range of scientists.

I think the paper, whether or not it finds a place in Nature Commun., needs to be rewritten to better reflect the key contribution. There needs to be some clarifications and some restructuring and rewording for the paper to better present its results. I think historically the authors have first found the results with monitoring, and that is why they insist in including it. Whereas, the key results for me is in column three of Table I (Eq. (1) without E).

Response: Thanks for the objective comments. We agree that from the view of practicality and performance of QKD, the case of without monitoring signal disturbance is more impressive. Therefore, we have revised the manuscript intensively, especially the results section, to focus more on the 'without-monitoring' case. Let us introduce some important revisions briefly.

1) The Fig.3 in previous version is removed. Thus, we do not simulate the key rate with monitoring signal disturbance vs channel loss now. Instead, the new Fig.3 is discussing the key rate without monitoring signal disturbance vs e_{mis} under typical

channel loss. With this new figure, we further analyze the performance of RRDPS in various optical misalignment conditions.

2) We add the case of $L=8$ to Tab. I. More discussions on this table are introduced.

More importantly, to further enrich the results of without monitoring, a new table (Tab. II) on maximum tolerable channel loss is added below Tab. I. With this new table, we clearly see that with the help of formula (1), RRDPS can be able to generate secret key under larger channel loss and L can be lowered obviously, compared to the original RRDPS. When $e_{\text{mis}} > 0.08$, the improved RRDPS can outperform BB84 in term of tolerable channel loss. Besides, one can see that the improvement due to formula (1) is very notable, while introducing formula (2), i.e. monitoring signal disturbance, leads further but very limited increase of tolerable channel loss. Thus, this figure makes sure that our results are very useful for the case of without monitoring.

3) We add an additional experiment without monitoring signal disturbance and decoy states for $L=3$. It is demonstrated that our improved RRDPS protocol with $L=3$ can generate secret key bits at the distance of 30 km without monitoring signal disturbance and decoy states. The experiment results of $L=3$ can illustrate the essential improvement of our modified RRDPS protocol in the case of without monitoring signal disturbance.

4) To focus more on small L cases, we add simulations for $L<10$ instead of $L=64$.

5) Some discussions and conclusions are reworded to focus more on 'without-monitoring' case.

For the completeness and significance of the security proof, we retain the theoretical result on monitoring signal disturbance, i.e. formula (2). For the purpose of verifying our theory, some results of simulations for the case with monitoring signal disturbance are retained.

I think the present manuscript has focused more on the 'without-monitoring' part of the result. We appreciate the Reviewer #2 for the valuable comments to help us improve our manuscript.

The bound on I_{AE} is different from I_{AE} itself. In many places the difference has been ignored and I_{AE} refers to its upper bound without being explicit.

Response: Corrections are made. And I_{AE}^U means the upper bound of I_{AE} now.

- I think on page 2, it should be de Finetti theorem (not defitti).

Response: Yes, it has been corrected.

- The notation and explanation on page 3 for the N-photon case is not clear.

Response: The notations and explanations are clearer now. Some examples are introduced to help readers understand the security proof well. However, for the throughout understanding of the security proof, it is better to read the supplementary file.

- On Pg 4, "tolerance of E" needs to be clearly defined.

Response: There is a maximum value E_{\max} of error rate E , which satisfies $I_{AB}=I_{AE}^U$ when $E = E_{\max}$. Obviously, if $E > E_{\max}$, $I_{AB}<I_{AE}^U$ and no secret key bits can be generated. Thus, E_{\max} is the maximum value of tolerable error rate of a QKD protocol. In the revised version, we discuss E_{\max} of different protocols. With the definition of E_{\max} , the related content is clear.

- e_{mis} needs to be clearly defined in such time/phase encoding techniques.

Response: In a phase-coding system, the error rate mainly stems from apparatus imperfections, such as interferometer misalignment and dark counts of SPD. The interferometer misalignment e_{mis} indicates the probability that an incoming photon hits an erroneous SPD due to interferometer misalignment. In a phase-coding system, e_{mis} depends on the visibility V of optical interference, and $e_{\text{mis}}=(1-V)/2$. In an ideal interferometer with $V=1$, two optical pulses with relative phases 0 and π always hit

different SPDs respectively. Thus one can deduce the relative phase, i.e. key bit in phase coding system, by observing which SPD clicks. However, due to the limited precision of interferometers fabrication or the environmental disturbance, e.g. the fluctuation of temperature, the lengths of the short arm and long arm of interferometer may drift. Then the visibility V may be lowered and higher e_{mis} is introduced. In RRDPS, the well matching of interferometers with variable delays is a challenging technique, thus e_{mis} may vary in different conditions.

A clear definition of e_{mis} has been added in the revised version.

- I think the pulse width is assumed constant for different values of L . This should be clarified in the text.

Response: Yes it is assumed constant for different values of L . We have clarified this in the revised version.

- The choice of $e_{\text{mis}} = 0.15$ must be justified by giving a practical scenario in which this is expected.

Response: Though e_{mis} in most of the reported QKD experiments can be kept to be small, i.e. $e_{\text{mis}} < 5\%$, it is still important to evaluate the performance of RRDPS in high e_{mis} region. There are three reasons accounted for the necessity of simulation with high e_{mis} .

1) Subtle and complicated active feedback techniques, e.g. phase-reference alignments and polarization controls, are indispensable to lower e_{mis} (Opt. Express 15, 16339(2011)). However, these techniques may be costly and even invalid in fast-changing environments. To improve the robustness of QKD system in various environments and alleviate its dependence on feedback techniques, QKD protocols inherently feasible in high e_{mis} scenarios are highly desired. Our simulations with high e_{mis} confirm that RRDPS is such a protocol that can be feasible in high e_{mis} scenarios and less dependent on active feedback setups.

2) The precision of interferometer fabrication is also essential to lower e_{mis} in phase-coding QKD systems. In RRDPS, this is much more challenging since the

interferometers with variable delays must be matching well. Thus, to alleviate QKD's demanding on this 'matching' technique, RRDPS in high e_{mis} scenarios are worth discussing.

3) Use of other high-dimensional degrees of freedom, e.g. orbital angular momentum (OAM) of photons, rather than time-bin, is a potential way to improve the key rate of RRDPS. Meanwhile, it is reported that its equivalent e_{mis} in QKD based on OAM can be typically larger than 10% (Proc. Natl. Acad. Sci. USA 112, 14197 (2015); arXiv: 1612.05195). Hence, simulations of RRDPS in high e_{mis} scenarios are relevant for future study. Our simulations with high e_{mis} confirm that RRDPS may be fit for the combination with OAM in future.

-Alternatively, if the authors want to show the advantage of their proof, they should find more realistic examples where by a choice of $L < 10$, they get some advantage out of their system.

Response: In Tab. II, we show that with our formula (1), the RRDPS with $L=7$ outperforms the original RRDPS with any L values in term of achievable distance. And the RRDPS with $L=8$ outperforms the BB84 in term of achievable distance, when e_{mis} is larger than 10%.

In Fig. 1, with our formula (1), the RRDPS with $L=8$ outperforms the original RRDPS with $L=16$ in terms of rate and reach.

These examples have demonstrated the advantages of our system.

- For me Fig. 3 does not have much to offer. We use a complicated system with $L=16$ plus monitoring, and we still do not outperform BB84+decoy.

Response: The old Fig.3 has been removed. Instead, the new Fig.3 is discussing the key rate without monitoring signal disturbance vs e_{mis} under typical channel loss. With this new figure, we further analyze the performance of RRDPS in various misalignment conditions. This new figure also shows that our improved RRDPS scheme with $L=8$ can work even when e_{mis} is larger than 11%.

- Pg 6, what kind of optimization VA does? is it for different distances?

Response: We apologize for this misleading description here. We use the intensity modulator to modulate different intensity to prepare signal, decoy and vacuum states. We just use the viable attenuator (VA) to adjust the mean photon number to the proper level.

There is no optimization for VA. The simulation for decoy states shows that difference between the optimal mean photon-numbers at different distances are not large. Thus for simplicity, we do not change the mean photon-numbers with different distances in our experiment. The related descriptions are corrected.

However, based on the simulations for RRDPS without monitoring and decoy states, its optimal mean photon-number is quite smaller than the one with monitoring and decoy states. Thus in the new experiment without monitoring signal disturbance and decoy states, we set a different mean photon-number from the cases using decoy states.

To Reviewer #3

We thank Reviewer #3 for the careful reading of our paper and helpful comments.

First point: Put results/Choice of protocol in context for the field.

The authors have done the barest minimum to address this point, in that I feel they just laundry list a bunch of other protocols. They make it clear that RRDPS does not need to measure the error rate, but do not say why that is helpful. Is it hard to measure this rate? I guess as a reader, I would still be unconvinced as to why I should use RRDPS.

Response: We are very grateful for this constructive comment. The introduction should be revised to present the differences and advantages of RRDPS better, just as required by the Reviewer #3.

Thus in the revised introduction, we introduce some representative QKD protocols, then discuss RRDPS and its notable character, i.e. it can run without monitoring signal disturbance. In following, we explain why without monitoring is an attractive feature.

There are several reasons for why RRDPS can become a very attractive protocol. In theory, the finding that QKD protocol may be secure even when monitoring signal disturbance is turned off sheds new light on how intrinsic randomness of quantum mechanics can be related to secure key distribution. In practice, the removal of monitoring signal disturbance means that we do not need to consider the statistical fluctuations in monitoring error rates and some other parameters, thus a better tolerance of finite-sized-key effects is expected. Especially, the finite-sized-key effects must be carefully considered in practice, since the fluctuations induced by environmental disturbance will lead to inaccurate statistical results or much more consumption for sampling. From the view of QKD engineering, the post-processing of QKD can be also simplified, since the random sampling and classical authenticated communications necessary in monitoring signal disturbance are all bypassed now.

Furthermore, according to the formula $I_{AE} \leq h_2(N/(L - 1))$, it's obvious that the information leakage will be deeply suppressed and positive key rate under higher error rate is expected when L becomes larger. Thus RRDPS is expected to have higher tolerance of error rate.

The introduction section has been revised intensively to explain clearly why RRDPS is a useful protocol.

Second point: The additional details in the methods section are greatly appreciated, however there should be more citations for where the formulas come from in this section.

Response: We have added more details to explain how to obtain the formulas. For example, let's explain here how we obtain the formula (3), which is the probability that Bob obtains one raw key bit per packet when single photon source is equipped.

Imagining that the delay value is r and each of the two SPDs opens one time-window to detect the incoming single photon, a single click is obviously occurred with the probability $(L - r)\eta/L + (1 - (L - r)\eta/L)2d(1 - d)$. The first item $(L - r)\eta/L$ corresponds to the probability that the single photon is not absorbed by channel or lost due to the noninterfering events in unbalanced Mach-Zehnder interferometers (Phys. Rev. Lett. **68**, 3121 (1992) and Rev. Mod. Phys. **74**, 145 (2002)), and then clicks a SPD. The second item means a dark count occurs while the single photon is absorbed by channel or lost due to the noninterfering events in interferometers.

Then considering that Bob actually opens $L - r$ time-windows and only retains the case that there is only one click among these $L - r$ time-windows, the probability that Bob obtains one raw key bit per packet is given by

$$Y_r = (1 - d)^{2(L-r)-1} \left(\frac{L-r}{L} \eta + \left(1 - \frac{L-r}{L} \eta \right) 2d(1 - d) \right).$$

I am very glad the authors were willing to upload their code. Unfortunately, I do not have a licence to Mathematica so I am unable to actually run or evaluate the code. That said, I would commend the authors for sharing the code because this is critical to good science and the review process.

Third point: The same response to point 2 applies here. I cannot run the code, as I do not have a licence to this software. I understand that the authors do not implement their own post-processing and this is fine, as long as this is stated and the formulas used to estimate the secret key rate are stated.

I unfortunately cannot make a recommendation on this paper, because I cannot actually evaluate the code, which supports large sections of the paper.

Response: We are sorry that the Reviewer #3 does not have a license of Mathematica to run our program. Unfortunately, we could not provide the simulation code using other programming tools at present. We hope we can provide the code in the future if condition permits.

Alternatively, we print the source code for our simulation as some supplementary (PDF) files in this resubmission, by which we hope the Reviewer #3 can evaluate our source code.

The “calculateIAE.pdf” file contains source code that defines the functions to calculate I_{AE} with different L and photon-number N . The “SPS.pdf” file contains the source code that simulates the secret key rate of RRDPS and BB84 when single photon source is equipped. The “noerrorate.pdf” file records the source code to simulate secret key rate versus channel loss without monitoring signal disturbance. The “ratevsemis.pdf” file records the source code to simulate secret key rate versus e_{mis} without monitoring signal disturbance. The file “forL=65.pdf” records the source code to calculate the secret key rate based on the already existing experiment with $L=65$.

REVIEWERS' COMMENTS:

Reviewer #2 (Remarks to the Author):

I believe the authors have addressed major points of concern in their revised manuscript, and I am happy to recommend publication if the following minor issues are also fixed:

- The paper still needs more proofreading and polishing, but I assume the editorial team can help with that. In particular, I would avoid using expressive terms such as dramatically, greatly, and so on.

- On Pg 2, line 1: It would be better if we specify classical "public-key" cryptography. Not all classical cryptography algorithms rely on computation complexity.

- Section I, last sentence: It is not clear from the sentence if the 140-km experiment has been done, and can, in principle, be done.

- Sec II, Sec A, para 1: The use of de Finetti theorem in distributed phase QKD protocols has not been straightforward as the symmetrization condition may not be held. Perhaps the authors need to be more cautious with this issue and better clarify why the use of this theorem is justified. Alternatively, the claims must be watered down.

- Sec II, Sec A, para 2: The way I understand $|i_1, i_2, i_3\rangle_{\text{odd}}$ notation is that in time slots i_1 , i_2 , and i_3 we have odd number of photons. But that does not specify a unique state, as the number of photons in each of these slots could be different. For instance, we can have one photon in i_1 but three photons in i_2 and i_3 , versus 3 photons in i_1 and i_2 and one photon in i_3 . Both cases may represent the case of $N=7$. The authors could perhaps clarify this issue by giving some examples.

- Pg 10, RRDPS with weak coherent source: Why do we assume resolving detectors are available to Bob? That makes the scheme less practical, which is against the motif for using coherent pulses. Is it just a matter of analytical convenience? is it a good approximation to the practical case where non-resolving detectors are in use? This needs to be addressed.

We thank Reviewer #2 for the helpful comments. The corresponding replies are given in blue as follows.

To Reviewer #2

I believe the authors have addressed major points of concern in their revised manuscript, and I am happy to recommend publication if the following minor issues are also fixed:

We thank Reviewer #2 for the careful reading of our paper and helpful comments.

- The paper still needs more proofreading and polishing, but I assume the editorial team can help with that. In particular, I would avoid using expressive terms such as dramatically, greatly, and so on.

The expressive terms mentioned above have been removed. The paper has been polished by Nature Research Editing Service. We hope that the English language in the text now meets the criteria for publication.

- On Pg 2, line 1: It would be better if we specify classical "public-key" cryptography. Not all classical cryptography algorithms rely on computation complexity.

This has been corrected.

- Section I, last sentence: It is not clear from the sentence if the 140-km experiment has been done, and can, in principle, be done.

The 140 km experiment has been performed here. We have clarified this point in the revised version.

- Sec II, Sec A, para 1: The use of de Finetti theorem in distributed phase QKD protocols has not been straightforward as the symmetrization condition may not be

held. Perhaps the authors need to be more cautious with this issue and better clarify why the use of this theorem is justified. Alternatively, the claims must be watered down.

Yes, we agree that the de Finetti theorem is not directly applicable in distributed phase QKD protocols. In Ref [Phys. Rev. Lett. 109, 260501 (2012)], it has been noted that this theorem holds by grouping the entire signal stream into blocks, so for distributed phase QKD, security against collective attacks on permutationally invariant blocks implies security against coherent attacks. In RRDPS, packets just corresponds to the permutationally invariant blocks. Thus, the de Finetti theorem is valid for the packets in RRDPS.

- Sec II, Sec A, para 2: The way I understand $|i_1, i_2, i_3\rangle_{\text{odd}}$ notation is that in time slots i_1 , i_2 , and i_3 we have odd number of photons. But that does not specify a unique state, as the number of photons in each of these slots could be different. For instance, we can have one photon in i_1 but three photons in i_2 and i_3 , versus 3 photons in i_1 and i_2 and one photon in i_3 . Both cases may represent the case of $N=7$. The authors could perhaps clarify this issue by giving some examples.

We apologize that this definition is not strict and clear.

Actually, $|i_1 i_2 \dots i_t\rangle_{\text{odd}}$ is a superposition of quantum states in which the photon-numbers in time-bins $i_1 i_2 \dots i_t$ are odd, while the photon-numbers in all other time-bins must be even. The strict form of $|i_1 i_2 \dots i_t\rangle_{\text{odd}}$ depends on the values of total photon-numbers N and the size of packet L . For example, if $N=3$ and $L=5$, nonnormalized $|1\rangle_{\text{odd}}$ will be $\sqrt{3}|1\rangle + (|2\rangle + |2\rangle + |3\rangle + |3\rangle + |4\rangle + |4\rangle + |5\rangle + |5\rangle) + |1\rangle + |1\rangle + |1\rangle$, where $|1\rangle + |2\rangle + |2\rangle$ means that there is one photon in the first time-bin and two photons in the second time-bin, and $|1\rangle + |1\rangle + |1\rangle$ means that all three photons occupy the first time-bin (see Supplementary Note 3 for the case of $N = 3$).

We have modified the main text and added the example of $|1\rangle_{\text{odd}}$.

- Pg 10, RRDPS with weak coherent source: Why do we assume resolving detectors are available to Bob? That makes the scheme less practical, which is against the motif for using coherent pulses. Is it just a matter of analytical convenience? is it a good approximation to the practical case where non-resolving detectors are in use? This needs to be addressed.

Yes, the assumption of the photon-number resolving detector is just for simplicity of simulation and convenience of analytical formulae.

The purpose of the simulation is to compare between the improved RRDPS and other protocols, but not the precise values of key rate or achieved distance. The assumption of the photon-number resolving detector is used for all protocols simulated here and thus does not impact the results of this comparison.

Although the security proofs in the manuscript and previous works [Nature 509, 475 (2014); New J. Phys 19, 033013 (2017); Phys. Rev. A. 95, 042301 (2017)] all assume that Alice may emit multi-photon state and Bob's measurement operator always applies to the single-photon state, the photon-number resolving detector is not necessary in practice. As proposed in the methods section of Ref. [Nat. Photon. 9, 827-831 (2015)], one can use the number of double-clicks of threshold SPDs to effectively estimate the number of times Bob receives a multi-photon packet, and then the potential information leakage due to multi-photon receiving events can be upper bounded and further eliminated by privacy amplification.

Therefore, we can safely assume a photon-number resolving detector, which is a good approximation to the practical case and does not affect the results of the comparison between the improved and original RRDPS.

The related text in the methods section has been revised.